



# An assessment of the impact of a nation-wide lockdown on air pollution – a remote sensing perspective over India

Mahesh Pathakoti[1]*, Aarathi Muppalla[2], Sayan Hazra[3], Mahalakshmi Dangeti[1], Raja Shekhar[2], Srinivasulu Jella[1], Sesha Sai Mullapudi[1], Prasad Andugulapati [2], Uma Vijayasundaram[3]

[1]Analytics and Modelling Division; Land and Atmospheric Physics Division; Earth and Climate Sciences Area, National Remote Sensing Centre (NRSC), Indian Space Research Organization (ISRO), Hyderabad-500037, India.
[2]Bhuvan Project Management and Software Evaluation Division, Bhuvan Geoportal and Data Dissemination Area, NRSC, ISRO, Hyderabad-500037, India.
[3]Department of Computer Science, School of Engineering & Technology, Pondicherry University, Chinna Kalapet, Kalapet, 
Puducherry-605014, India

*Correspondence to*: Mahesh Pathakoti (mahi952@gmail.com)

**Abstract.** The nation-wide lockdown imposed over India from 25th March 2020 onwards, in response to the COVID-19
pandemic, placed severe restrictions upon the industrial and transport sectors, which together form a significant chunk of anthropogenic emissions of pollutants into the atmosphere. Atmospheric concentrations of Nitrogen dioxide ($NO_2$), carbon monoxide (CO) and aerosol optical depth (AOD) for lockdown and pre-lockdown periods were investigated with observations from Aura/ OMI, Terra/ MOPITT, Sentinel-5p/TROPOMI and Aqua-Terra/ MODIS satellite sensors. Mean $NO_2$ levels over India during the lockdown period showed a dip of 17% as compared to pre-lockdown period and a decrease of 18% against
the 5-year average. Over New Delhi in particular, there was a sharp decrease of 62% in $NO_2$ levels as compared to 2019 and a decline by 54% relative to the preceding 5-year period (2015 - 2019). Aerosol levels reduced over the country by ~24% from the 5-year mean levels, with a marked reduction over the Indo-Gangetic plains region. An increase in CO levels was noticeable, probably due to its longer life-time as compared to $NO_2$ and aerosols. This study also reports the rate of change of $NO_2$, CO and AOD, indicating increase/decrease in pollutant emissions over the different states of India.

**Keywords:** COVID-19, lockdown, remote sensing, $NO_2$, CO, AOD

## 1 Introduction

Following the outbreak of the novel corona virus (COVID-19) and its declaration by the World Health Organization (WHO) to be a public health emergency of international concern, several countries across the globe imposed national lockdowns to contain the pandemic (Tian et al., 2020). India confirmed its first COVID-19 case on 30th January 2020 with an exponential
increase to 360 cases by 22nd March 2020. In an attempt to restrict the pandemic, the Indian government called for a 'Janata Curfew' on 22nd March 2020, followed by imposition of a nation-wide lockdown in phase I (25th March – 14th April) and phase II (15th April – 3rd May) which has been further extended (Gettleman and Schultz, Kai, 2020; UN news, 2020). Under the lockdown, 1.30 billion citizens were advised to stay in-doors, transport and industrial production suspended and only essential





services were permitted. During this 40-day period, economic activities were greatly affected and there was a fall in net energy

consumption by 30% (https://www.ppac.gov.in/). The prolonged cessation of industrial and vehicular activities during COVID-19 lockdown in china caused significant reduction in air pollutants over its 3 cities (Xu et al., 2020; Fan et al., 2020).

In recent times air pollution has arisen as an environmental issue of major concern worldwide, particularly for developing countries, and may extend from local to global scale (Fang et al., 2009). Emissions of primary pollutants comprising particulate

matter (aerosols) and gases (nitrogen dioxide / $NO_2$ and carbon monoxide / CO) play a vital role on environment and human health (Xu et al., 2020). Earth-atmospheric radiation budget is greatly affected by the aerosols through scattering and absorption of incoming solar radiation, which further influences the formation of clouds and precipitation (Ramachandran et al., 2013). The size of aerosols can affect formation of clouds and thus aerosols can influence the Indian monsoon (David et al., 2018). Exposure to $NO_2$ has been correlated with an increased rate of morbidity and subsequently increased rate of mortality

(WHO, 2013). Global emissions of $NO_x$ (NO, $NO_2$) are primarily due to anthropogenic activities such as combustion of fossil fuel and industrial activities, while natural sources include soils and lightning. Among these, combustion is the principal process through which trace gases and aerosols are emitted into the atmosphere. Rapid industrialization and exhaust from vehicles are major sources of air pollution in urban regions. The ambient air quality is largely determined by the abundance of these primary pollutants in the atmosphere (Nishanth et al., 2014). Increase in the concentration levels of trace gases has been

a challenging environmental issue in urban and industrial areas. According to the National Ambient Air Quality Monitoring Program, emissions of $NO_2$ over the Indian region are second only to particulate matter emissions and are higher than sulfur dioxide and CO (Gaur et al., 2014). CO acts as both a greenhouse gas and a precursor for ozone chemistry in the lower atmosphere. During the lockdown period in India, ground-based in-situ analysis evidenced strong decline in air pollutants (Mohato and Ghosh, 2020).


In spite of the various nationwide lockdowns which may have led to suspension of ground-based measurements, remote sensing data offered a reliable and accurate method to measure air pollution with un-interrupted observations. In the present study we attempted to assess the impact of lockdown on air quality over India by examining remotely sensed $NO_2$, CO, aerosol optical depth data from Aura/Ozone Monitoring Instrument (OMI), Terra/ Measurements of Pollution In the Troposphere (MOPITT),

Sentinel-5p/Tropospheric Monitoring Instrument (TROPOMI) and Aqua-Terra/Moderate Resolution Imaging Spectroradiometer (MODIS) respectively. The spatio-temporal variations in these pollutants are examined and short-term climate change due to short-term climate forcing agents during the lockdown period is investigated.



## 2 Materials and Methods

| Parameter | Data source | Resolution (km) | Data source |
|-----------|-------------|-----------------|-------------|
| NO₂ | Aura/OMI | 25 km | https://earthdata.nasa.gov/ |
| CO | Terra/MOPITT | 100 km | https://earthdata.nasa.gov/ |
|    | Sentinel5p/TROPOMI | 10 km | |
| AOD | Aqua-Terra/MODIS | 100 km | https://giovanni.gsfc.nasa.gov/giovanni |
| *Reference Period: March-May (2015 – 2020); Lockdown Period: 25 March – 3 May 2020* | | | |

**Table 1**. Data resources

AOD, CO and $NO_2$ observations made by various satellite-based sensors as per Table 1 have been used for this study. $NO_2$ data at 25km spatial resolution was obtained from the daily level 3 $NO_2$ product from Aura/Ozone Monitoring Instrument (OMI). Level 2 CO data (version 8.0) at 1°×1° spatial resolution from the Terra/Measurement of Pollution In the Troposphere (MOPITT) sensor and at 7km resolution from the Sentinel-5p/Tropospheric Monitoring Instrument (TROPOMI) respectively, for the period between March 2015 - 16 February 2020 for the former and the subsequent period from the latter sensor. The TROPOMI instrument on-board Sentinel-5p, launched on 13th October 2017, measures at high spatial resolution the total columnar concentrations of CO, ozone, sulphur dioxide and formaldehyde. Level 3 Aerosol Optical Depth (AOD) observations from the Moderate Resolution Imaging Spectroradiometer (MODIS) instrument on-board Terra (MOD08_M3_v6 --Aerosol Optical Depth 550 nm, Deep Blue, Land-only) and Aqua(MYD08_M3_v6 -- Aerosol Optical Depth 550 nm (Deep Blue, Land-only ) satellites at 10:30 and 13:30 local time have been used to investigate the aerosol loading over the Indian region. Details about OMI onboard Aura and MODIS onboard Aqua/Terra are explained by Li et al. (2020).

The COVID-19 lockdown over India was implemented in several phases, viz. phase I (21 days from 25th March to 14th April 2020), phase II (19 days from 15th April to 3rd May 2020), phase III (14 days from 4th May to 17th May 2020) and phase IV (14 days 18th May to 31st May 2020). The present study focuses on the air quality over India, its individual states, and state capitals during the lockdown period. Analysis of satellite based observations of $NO_2$, CO and AOD was carried out for lockdown period as well as pre-lockdown period from 2015-2020. Short-term climatological mean was computed during 2015-2020 for the months corresponding to lockdown period to assess the temporal changes of pollutants in the atmosphere. We have focused our analysis for the first two phases of lockdown in which the industrial and transport sectors were brought to a near standstill. Figure 1 shows the data handling and execution strategy followed in this study. The regional increase/decrease in pollutant emissions over the country and individual states were analyzed.





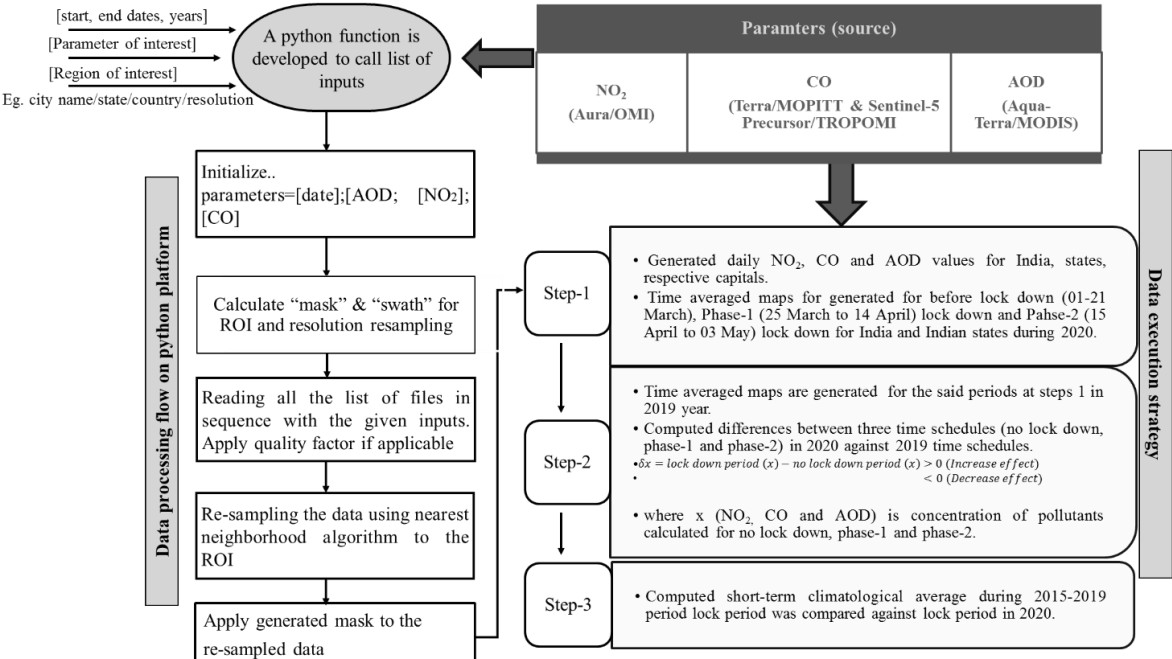

90         **Figure 1.** Data handling and processing methodology

## 3 Results and Discussion

### 3.1 Impact of lock down on NO₂ variations over Indian region

Tropospheric columnar $NO_2$ product from Aura/OMI which is 30% cloud screened and available at 25km spatial resolution has been analyzed for 2015-2020. Temporally averaged concentrations of $NO_2$ for the duration of lockdown period (25th March

95 to 3rd May 2020), phase I and phase II of lockdown, and the corresponding period of the previous year (2019) are shown in Figure 2a-c, along with the differences in concentration levels between different periods. In 2019, noticeably high concentrations of $NO_2$ were there over Delhi and eastern part of India. In sharp contrast to this, much lower levels of $NO_2$ over New Delhi, and a significant reduction in ambient levels throughout the country are observed during 2020 lockdown period. The mean $NO_2$ concentration over the country during the entire duration of lockdown, phase-I and phase-II period in 2020

100 (2019) are $2.01 \times 10^{15}$ ($2.42 \times 10^{15}$) molecules cm⁻², $2.03 \times 10^{15}$ ($2.45 \times 10^{15}$) molecules cm⁻² and $1.99 \times 10^{15}$ ($2.40 \times 10^{15}$) molecules cm⁻² respectively, with a reduction of 17 % during the lockdown period as compared to the previous year. However, an increase in $NO_2$ levels is observed in the north, north-east and western parts of India, which may possibly be due to seasonal change. Figure 2d shows the weekly variations of $NO_2$ during the lockdown period. $NO_2$ life time is shorter at day time due to reactive photochemical processes in presence of sunlight and longer at night (Richter et al., 2004). Satellite measurements from

105 Aura/OMI are able to detect build up and removal of $NO_2$ emissions during the lockdown period, as evidenced by Figure 2d.



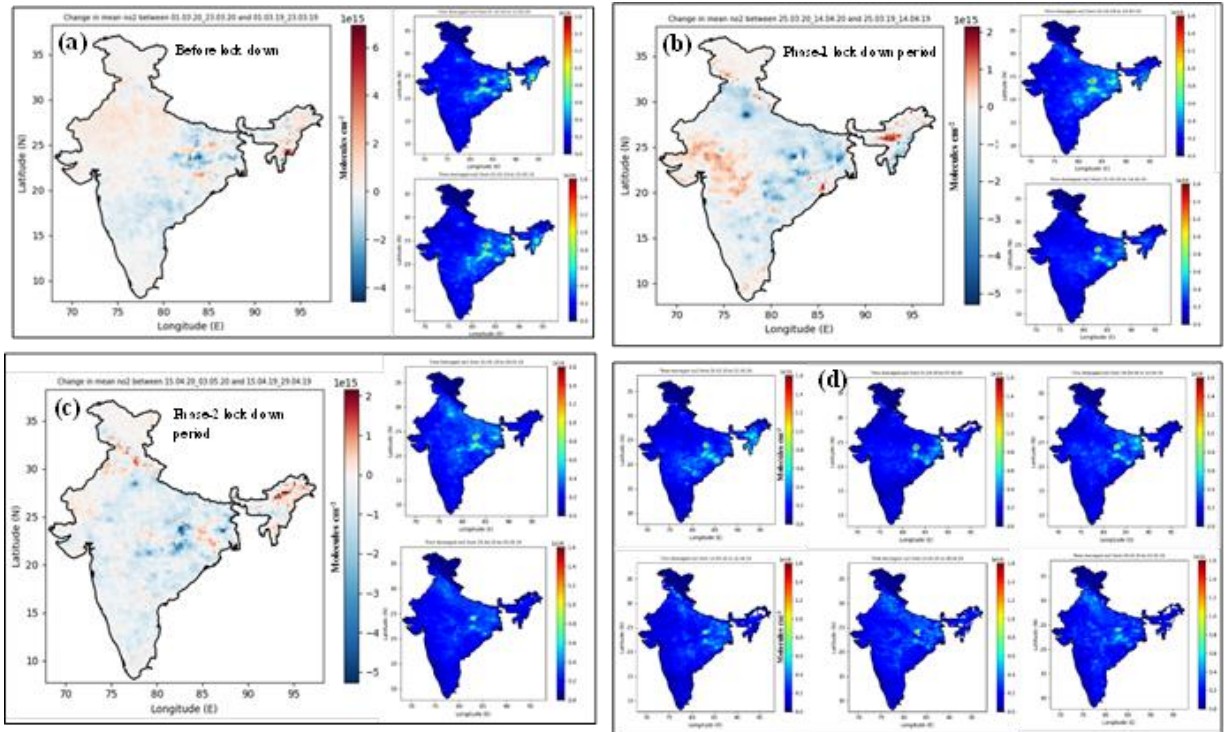

**Figure 2.** Time averaged $NO_2$ concentration and their difference maps in 2020 and 2019 **a)** during total lock down period **b)** phase-1 lock down **c)** phase-II lock down, **d)** weekly variation in total lock period. The color bar on spatial plots shows column $NO_2$ values.

### 3.1.1 Short-term climatological variations in $NO_2$ due to lock down

Time series analysis of $NO_2$ was carried out during 2015-2020 for the months corresponding to lockdown period (Figure 3a-b) to understand the temporal variations in $NO_2$ levels in the atmosphere. A smoothing function with span of 7 days was used for better visualization of patterns/trends in $NO_2$ levels in figure 3a-b, with red (green) bars indicating elevated (lowered) $NO_2$ levels in 2020 relative to 2019 and 2015-2019. Strong reductions of $NO_2$ were reported in many regions of Spain. A substantial part of these reductions is obviously due to decreased local and regional anthropogenic emissions (Petetin et al., 2020). Due to low emissions of $NO_2$ during lockdown period, it is observed that the ambient concentrations did not rise back to 2019 levels. The short-term climatological mean (2015-2019) $NO_2$ levels are contrasted against lockdown period in Figure 3b. Due to the strict lockdown conditions, ambient $NO_2$ levels from 1st March – 30th April 2020 were ~$2.08 \times 10^{15}$ molecules cm$^{-2}$ which was nearly 14.5% lower than the short-term climatological average. Bar plots of $NO_2$ levels during 2015-2020 and the rate of change (RoC) in $NO_2$ levels against the short-term climatological mean (Figure 3c-d) indicate the impact of lockdown on $NO_2$ emissions over Indian region. The RoC is extremely important in weather and climatological studies because it allows understanding and predicting the trends/patterns in climatic parameters. RoC is used to describe the percentage change in a





parameter over a defined period of time and it represents the rate of acceleration of the parameter. There is a clearly observable lowering in $NO_2$ levels relative to the short-term climatological mean by 12% for the pre-lockdown period, 18% and 8% for

the phase I and phase II lockdown periods, respectively.

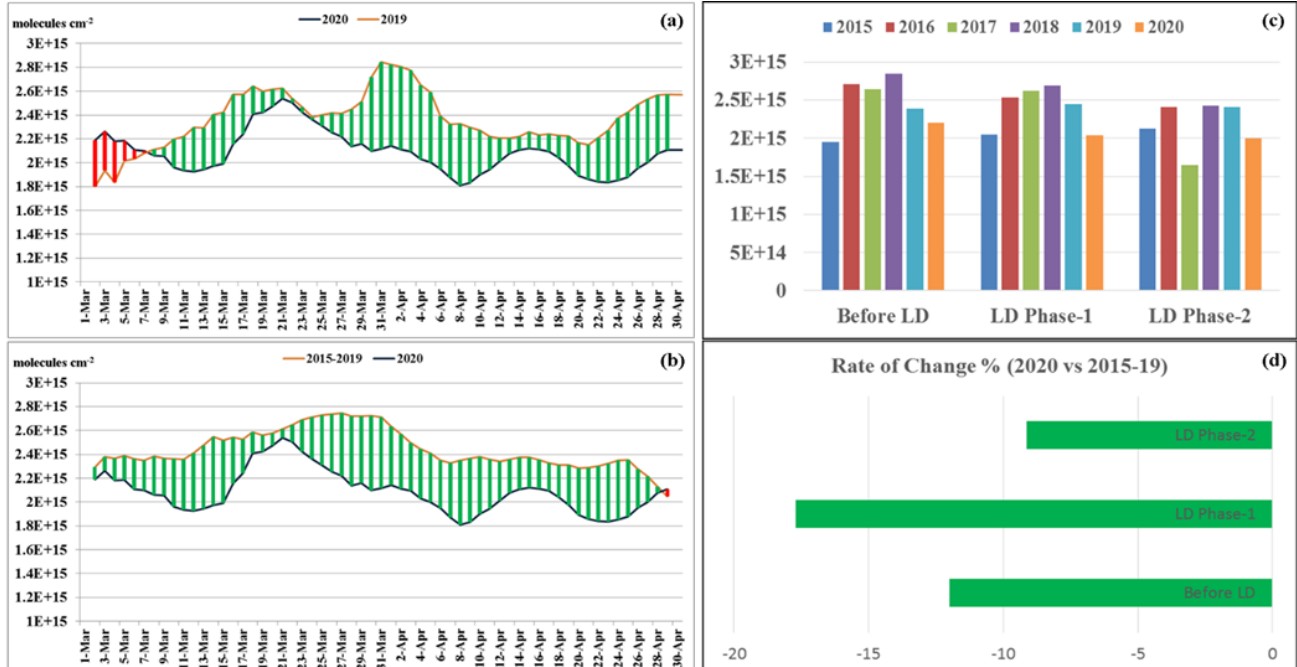

**Figure 3.** Moving average time series analysis of $NO_2$ during **a)** 2019-2020; **b)** short-term climatological mean of $NO_2$ (2015-2019) vs. 2020 (March-April months) **c)** Variations of $NO_2$ before LD and different phases of LD and **d)** Rate of change of $NO_2$ during 2020 w.r.t. 2015-2019 period.

**3.2  Impact of lock down on CO  variations over Indian region**

Tropospheric CO data was obtained from Terra/MOPITT (2015-2019) and Sentinel-5p/TROPOMI (2020) respectively. The mean CO levels over Indian region during the pre-lockdown and lockdown periods, comparison of CO levels during lockdown period against the 5-year short-term climatological mean for 2015-2020 periods were studied to assess the short-term climatic effect due to COVID-19 lock down. Figure 4(a-c) shows mean concentrations of tropospheric CO over the Indian region during

pre-lockdown phase I and phase II lockdown periods. During the pre-lockdown period (1-10[th] March 2020), CO levels were higher ($\delta = 0.04 \times 10^{17}$ molecules $cm^{-2}$) as compared to 2019 by ~2%, which is expected due to anthropogenic activities. During phase I (mean = $2.50 \times 10^{18}$ molecules $cm^{-2}$) there is a decrease in CO levels over north and south India as compared to 2019 (mean = $2.28 \times 10^{18}$ molecules $cm^{-2}$). However, the mean CO levels are actually higher during the phase II of lockdown. This indicates that the longer life time of CO molecules in the atmosphere (~1-2 months) relative to $NO_2$ has resulted in not much





variation in atmospheric CO levels. The weekly variability of CO plotted using TROPOMI data, indicates spatial variation in CO emissions during the lock down period as shown in figure 4d.

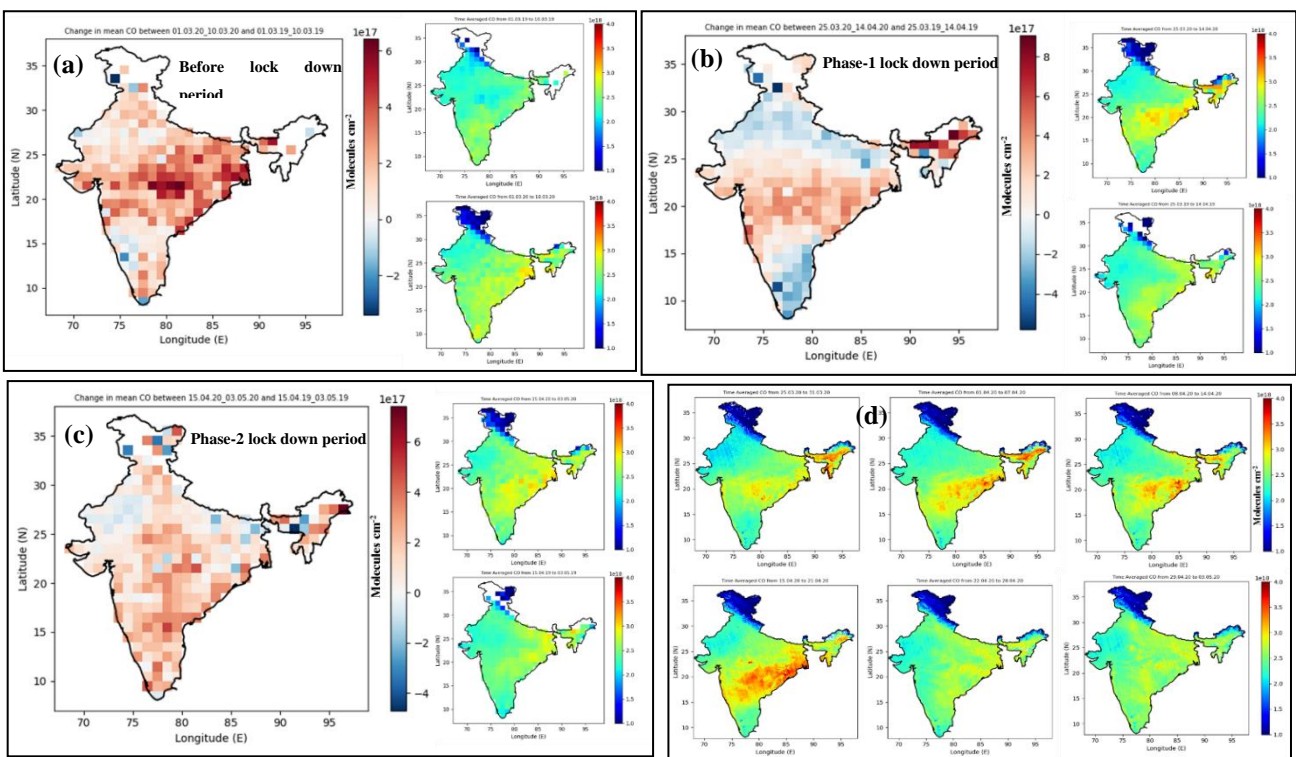

**Figure 4**. Terra/MOPITT during 2019 and Sentinel-5p/TROPOMI during lock period derived **a)** Time averaged CO concentration and their difference maps in 2020 and 2019 before lock down period **b)** phase-1 lock down **c)** phase-2 lock down, **d)** weekly variation in total lock period during 2020.

The 5-year (2015-2019) mean CO emissions against 2020 during pre-lockdown, phase-1 and phase-2 lock periods show an increase of CO indicating duration of the (N = 40 days) lock down did not capture instant and short-term climatic change (Figure is not shown here). Rate of change during pre-lock down, phase-1 and phase-2 lockdown in 2020 w.r.t to 2015-2019 are -2 %, 9 % and 8 % respectively.

### 3.3 Impact of lock down on variability of AOD over Indian region

Besides greenhouse gases effect on Earth's climate, aerosols in the lower atmosphere also alters the earth's climate by the process of scattering and absorbing of incoming solar radiation, which also helps in altering cloud properties (Boucher et al., 2013). We have used $AOD_{550}$ products from Terra-Aqua/MODIS which are generated for 10:30 and 13:30 local time respectively. As we observed similar spatial variation of $AOD_{550}$ from both Terra-Aqua/MODIS, only Aqua/MODIS derived $AOD_{550}$ is shown here (Figure 4). $AOD_{550}$ levels over the Indian region for 2019, 2020 and the difference in $AOD_{550}$ for both





years for pre-lockdown period is depicted in Figure 5a. During this period, the $AOD_{550}$ levels for 2020 (0.39) over the Indian region were higher than for 2019 (0.35). The relatively higher aerosol levels in 2020 over Indo-Gangetic Plain (IGP) and central India during the pre-lockdown period could be due to seasonal loading of aerosols over the Indian region and pre-

monsoon western dusty winds, which elevates $AOD_{550}$ concentration (David et al., 2018). The maximum $AOD_{550}$ over Indian region varied between 0.8 to 1.0 during the lockdown period with a noticeable reduction in $AOD_{550}$ over the IGP region. Spatial maps of $AOD_{550}$ for 2019 indicate high levels of aerosol loading over the densely populated IGP due to the high industrial emissions in this area. The formation and removal of aerosols is influenced by emission sources, prevailing meteorological conditions and wet removal in monsoon season (Babu et al., 2013). Overall, a significant lowering of aerosols from the

atmosphere is observed over the Indian region during the lockdown period. Figure 5d show weekly $AOD_{550}$ variations during 40 days lock down period. First 3 weeks of lock down $AOD_{550}$ shows strong reduction in north and IGP regions whereas subsequent weeks show reduction in central India. Thus, timestamp analysis show India lock down had strong impact on aerosol loading over India which explains the significance of anthropogenic emissions on aerosol loading over India.

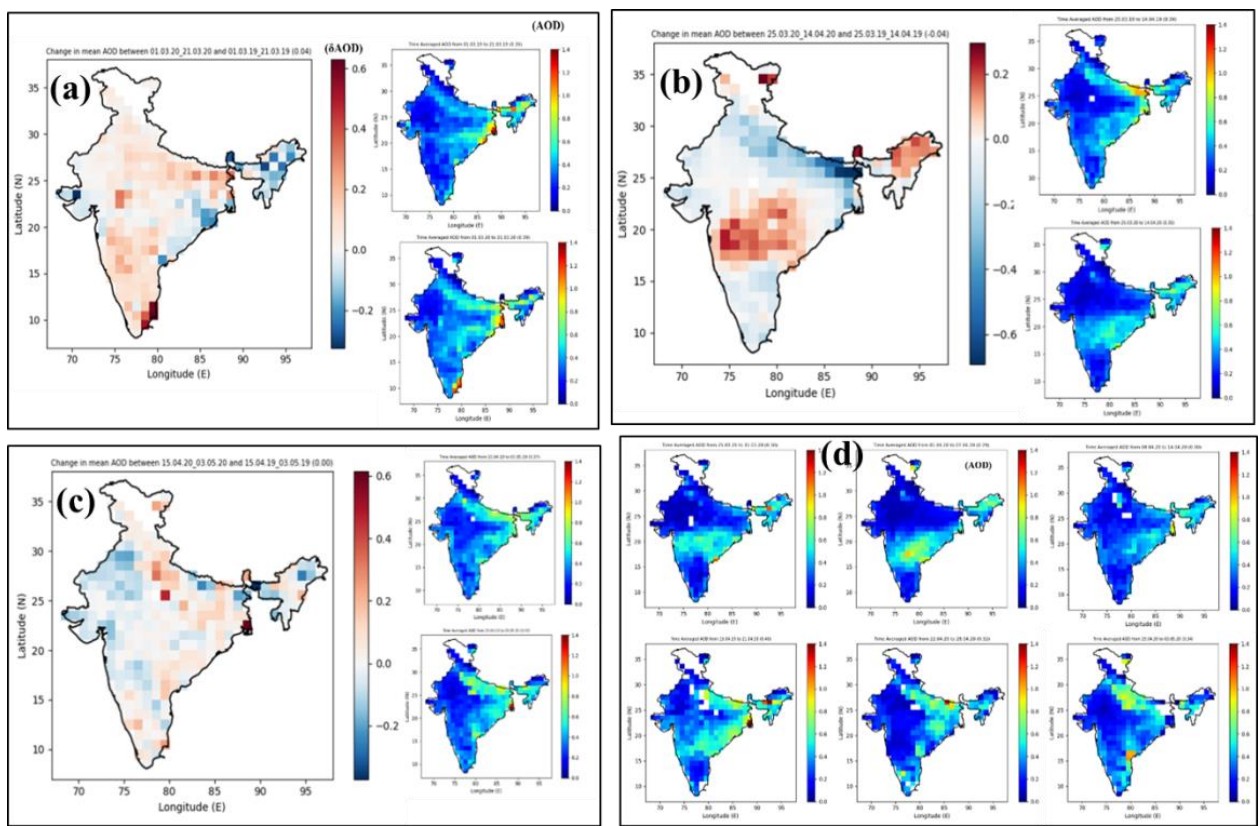

**Figure 5**. Aqua/MODIS $AOD_{550}$ **a)** Time averaged $AOD_{550}$ concentration and their difference maps in 2020 and 2019 before lock down period **b)** phase-1 lock down **c)** phase-2 lock down, **d)** weekly variation in total lock period during 2020.



### 3.3.1 Short-term climatological variation of AOD due to lock down

Aerosol optical depth is one of the important short-term climatic forcing agents along with long lived greenhouse gases namely
carbon dioxide ($CO_2$), methane ($CH_4$), water vapor ($H_2O$) and nitrous oxide ($N_2O$). A 7 day smoothing average filter was
applied on $AOD_{550}$ time series data as discussed in section 3.1.1. Figure 6a-d show 7-day averaged time series analysis of
$AOD_{550}$ levels over Indian region during 2019-2020 and contrast the $AOD_{550}$ levels during the lockdown period against the
short-term climatological mean of 2015-2019. The bar plots in Figure 6 indicate the seasonal dispersion of $AOD_{550}$ and
influence of other factors. Analysis of $AOD_{550}$ for pre-lockdown, phase I and phase II of lockdown period (compared to
corresponding period in 2019) indicated a high $AOD_{550}$ in pre-lockdown period followed by sharp drop during the lockdown
period, possibly due to decrease in anthropogenic activities. The Rate of change in $AOD_{550}$ was computed (figure 6e-f) to
understand the effect of short-term climatological mean $AOD_{550}$ over lock period $AOD_{550}$ in 2020. A RoC of 5 % positive
(negative) was observed before LD (red color), as measured by the Terra/MODIS (Aqua/MODIS) which shows poor
correlation during this period. However, strong reduction of 24-25 % in phase-1 and 10-12 % in phase-2 lock period was
observed against 5-year mean $AOD_{550}$. Thus, lock down helped to improve the air quality in forthcoming days. Aerosol
properties extend feasible support for modulating the cloud properties, which subsequently helps amount of precipitation
(Boucher et al., 2013). A study over the Indian sub-continent during Cloud Aerosol Interaction and Precipitation Enhancement
Experiment (CAIPEEX) in 2009 and 2010 was reported that a polluted cloud has more number of smaller drops, which causes
reduction in precipitation by the cloud (Konwar et al., 2012). Studying and quantifying the impact of aerosols on precipitation
is still a thrust area of research (Myhre et al., 2013). As precipitation is pre-dominantly influenced by the availability of
moisture and cloudiness, this lockdown has significantly altered aerosol loading over the Indian region, we hypothesize the
Indian summer monsoon of 2020 may improve as compared to the preceding year. However, strong evidences are not explored
in the present study.

**Figure 6. a)** Moving average time series analysis of $AOD_{550}$ measured by Terra-Aqua/MODIS) Terra/MODIS 2019-2020; **b)** short-term climatological mean of $AOD_{550}$ (2015-2019) vs. 2020; **c)** time series $AOD_{550}$ measured by Aqua/MODIS during 2019-2020; **d)** short-term climatological mean of $AOD_{550}$ (2015-2019) vs. 2020; **e)** Variations of Terra/MODIS measured $AOD_{550}$ before LD and different phases of LD and respective RoC; **f)** Variations of Aqua/MODIS measured $AOD_{550}$ before LD and different phases of LD and respective RoC during 2015-2020 period.

### 3.4 State wise rate of Change of $NO_2$, CO and AOD during lock down w.r.t. 2015-2019

Figure 7a-7d show state-wise RoC computed for $NO_2$, CO, $AOD_{550}$ at 10:30 and AOD 13:30 LT during lock period w.r.t 2015-2019 (N = 5 years) period. Positive (dark blue above zero) indicates increase of pollutants during the lock period when





compared to 2015-2019 mean emissions of $NO_2$, CO and $AOD_{550}$. Negative (light blue below zero) indicates decrease of $NO_2$,
CO and $AOD_{550}$ emissions w.r.t 2015-2019 values. Top 10 states with increase and decrease of these pollutants are listed in
the table 2.

**Table 2.** List of top 10 states of India with decrease (↓) and increase (↑) of $NO_2$, CO and $AOD_{550}$ w.r.t short-term climatological
mean.

| States ↓ (↑) –$NO_2$ | ROC % ↓ (↑) | States ↓(↑)-CO | ROC (%) ↓(↑) |
|---|---|---|---|
| New Delhi (Arunachal Pradesh) | -54.31 (7.72) | Jammu & Kashmir (Odisha) | -33.17 (15.52) |
| Sikkim (Tripura) | -33.20 (7.55) | Himachal Pradesh (Goa) | -28.09 (14.58) |
| Haryana (Meghalaya) | -31.37 (5.35) | Uttarakhand (Maharashtra) | -24.79 (13.98) |
| Uttarakhand (Assam) | -30.32 (2.21) | Sikkim (Andhra Pradesh) | -24.04 (12.46) |
| Punjab (Nagaland) | -29.23 (-0.18) | Manipur (Telangana) | -12.25 (11.35) |
| Goa (Jammu) | -28.91 (-3.48) | Arunachal Pradesh (Gujarat) | -11.35 (8.91) |
| Andhra Pradesh (Rajasthan) | -28.17 (-5.82) | Mizoram (Tripura) | -8.83 (6.7) |
| Jharkhand (Manipur) | -24.39 (-6.53) | Nagaland (West Bengal) | -6.05(5.94) |
| Karnataka (Mizoram) | -22.95 (-7.77) | Punjab (Kerala) | -1.07 (5.49) |
| Uttar Pradesh (Gujarat) | -21.96 (-9.28) | Rajasthan (Tamilnadu) | -0.07 (4.49) |
| **States ↓(↑)- Terra/MODIS-AOD** | **ROC (%) ↓(↑)** | **States ↓(↑)-Aqua/MODIS-AOD** | **ROC (%) ↓(↑)** |
| Uttarakhand (Sikkim) | -41.20 (22.11) | Haryana (Sikkim) | -45.48 (59.52) |
| Rajasthan (Arunachal Pradesh) | -40.44 (9.87) | Gujarat (Maharashtra) | -38.07(15.10) |
| Bihar (Telangana) | -39.36 (8.81) | Punjab (Arunachal Pradesh) | -37.15 (9.59) |
| Haryana (Chhattisgarh) | -38.24 (8.67) | Uttarakhand (Chhattisgarh) | -35.12(5.46) |
| Punjab (Jammu & Kashmir) | -36.04 (6.74) | Rajasthan (Telangana ) | -34.53(2.85) |
| Himachal Pradesh (Maharashtra) | -33.09 (6.73) | Tripura (Jammu & Kashmir) | -34.21(2.51) |
| West Bengal (Odisha) | -31.50 (-3.80) | West Bengal (Odisha) | -33.13(1.16) |
| Puducherry (Assam) | -29.75 (-9.95) | Kerala (Madhya Pradesh) | -31.97(-10.31) |
| Tripura (Andhra Pradesh) | -29.45 (-10.78) | Puducherry (Karnataka) | -30.97(-14.91) |
| Uttar Pradesh (Nagaland) | -29.41 (-14.17) | Uttar Pradesh (Goa) | -27.69(-16.06) |



**Figure 7.** Sate wise rate of Change computed before, phase-1, phase-2 and total lock period **a)** NO₂; **b)** Terra/MOPITT and Sentinel-5p TROPOMI derived CO; **c)** Terra/MODIS derived AOD$_{550}$ and **d)** Aqua/MODIS derived AOD$_{550}$





Satellite derived NO$_2$ levels over New Delhi reduced during the total lock period by 54.31 % w.r.t.to 2015-2019 and has
sharply declined by 62 % w.r.t 2019 levels. This reduction is consistent with the exhaustive lockdown effect such as reduction
of anthropogenic activities, transportation, shutdown of industries and power plants.  In Jammu and Kashmir State, CO levels
decreased by 33 % in lock period when compared to 2015-2019 mean CO emissions. This probably could be due to less sources
of emissions in this region. The Aqua-Terra/MODIS derived AOD show 80 % strong correlation of with decreasing states.
Terra/MODIS derived AOD show 41 % decline over Uttarakhand and Aqua/MODIS show 48 % decrease over Haryana states
respectively. The NO$_2$ and AOD show strong reduction of emission levels during the lock period compared to 2015-2019,
indicating reduction of similar sources of these pollutants. Blue color indicates decrease of NO$_2$, CO and AOD over Punjab
and Uttarakhand states during the lock period. Study depicts that India's exhaustive lock down being a world's largest
lockdown significantly eased air pollution in the country.

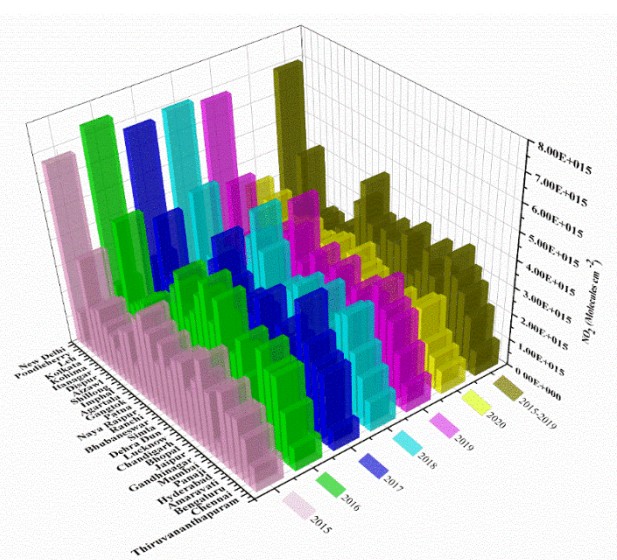

**Table 3**. List of Indian cities with sharp decline
of NO$_2$ emissions

| City name | ROC (%) w.r.t 2015-2019 ↓ | ROC (%) w.r.t 2019 ↓ |
|---|---|---|
| New Delhi | 54 | 62 |
| Bengaluru | 43 | 51 |
| Chennai | 41 | 48 |
| Chandigarh | 39 | 55 |
| Naya Raipur | 36 | 41 |
| Srinagar & JK | 36 | 20 |
| Mumbai | 35 | 44 |
| Gandhinagar | 33 | 31 |
| Hyderabad | 30 | 40 |
| Panaji | 29 | 23 |

**Figure 8.** NO$_2$ levels during 2015-2020 during lock down period

With the advantage of Aura/OMI spatial resolution of 25 km, city level NO$_2$ level variations are studied during 2015-2020
period as shown in figure 8. Lock down period of 25 March to 03 May days were considered for 2015-2020 years to study the
spatial variations in NO$_2$ levels. During all the years of 2020 lock down period, New Delhi shows high concentration of NO$_2$
compared to other state capitals. This could be due to population density of the city and sources of emissions including
anthropogenic activities. Under the lock down period conditions, NO$_2$ levels highly dropped over country capital city, New
Delhi by 62 % w.r.t 2019 and 54 % w.r.t time averaged NO$_2$ during 2015-2019. New Delhi is being known for one of the
highest polluted cities in the world. However, 40 days of exhaustive lock down eased air pollution over New Delhi even w.r.t





short-term climatological mean of $NO_2$, which indicates the impact of the India lockdown in connection with the COVID-19 resulted in enhanced air quality over the country. Table 3 shows major cities of India and their RoC in dropped $NO_2$ levels during lockdown period in 2020 against 2019 and short-term climatological mean.

## 4 Conclusions

Present study carried out analysis on air pollution in connection with the world's largest lock down imposed by India to contain the spread of COVID-19. The lockdown was extended over several phases viz., phase I (21 days from 25 March to 14 April 2020), phase II (19 days from 15 April to 3 May 2020), phase III (14 days from 4 May to 17 May 2020) and phase IV (14 days from 18 May to 31 May 2020). However, the lockdown was near total only in phase I and II, with the near total shutdown of industrial and transport sectors. We used satellite-based observations of tropospheric $NO_2$, CO and aerosol optical depth and analyzed the pollutant concentrations during the period of lockdown against the same period of the preceding year and contrasted against the short-term climatological mean for the 5 year period from 2015- 2019. Results of the study show strong reduction of $NO_2$ and AOD over the country, state, and cities whereas mixed variation observed in CO emissions. $NO_2$ levels over the Indian region decreased by 17 %. The 5-year short-term climatological mean of $NO_2$ during 2015-2019 shows 12 % decrease when compared to pre-lockdown and 18 % during phase-1 lock period, respectively. Aerosols / particulate matter content of the atmosphere, as quantified by aerosol optical depth (AOD), decreased over high emission regions (IGP) during the lockdown period. The short-term climatological mean of AOD showed strong reduction (24-25 %) during lock period, which is a positive indicator for the radiation budget and forthcoming seasonal rains in India. During lock period, CO has not turned up positive sign for air pollution, which could be due to longer lifetime compared to $NO_2$. Rate of change of $NO_2$, CO and AOD for each state was computed during lock period and compared against respective 5-year averaged emissions. Northern and eastern part of the country showed strong decrease of emissions due to lock down. Among the metropolitan cities of India, the capital city of New Delhi has shown greatly enhanced air quality during the lockdown period. Thus, exhaustive lock down due to outbreak of COVID-19 has led to improvement in the air quality and short-term climatic effects over India.

**Acknowledgement**

Authors sincerely thank Shri Santanu Chowdhury, Director NRSC for his support and encouragement for carrying out this study. We greatly acknowledge Earth data and Giovanni data web portal for providing the free access to the Aura/OMI, Terra/MOPITT, Sentinel-5p/TROPOMI and Aqua-Terra/MODIS satellites data. We thank Mr. S.V.S. Sai Krishna, Scientist, ECSA, NRSC and Dr.P. Raja, Principle Scientist, Indian Institute of Soil and Water Conservation-Indian Council of Agriculture Research, Ooty, India for reviewing the manuscript. Authors thank Ms. A.L. Kanchana, Scientist, ECSA, NRSC for helping with the data sets.



**Conflicts of Interest**

The authors declare no conflict of interest.

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
