# Peer review of "An assessment of the impact of a nation-wide lockdown on air pollution – a remote sensing perspective over India"

_Atmospheric Chemistry and Physics, 2020_

## Referee Comment (RC1) · Anonymous Referee #1 · 15 Jul 2020

This paper examines NO2 and CO concentrations and aerosol optical depth over India as observed by satellite remote sensing instruments before and during COVID-19-related lockdown restrictions in the country. The authors conclude that changes in emissions due to lockdown restrictions are responsible for the observed changes in air quality. While this is an interesting and timely topic of discussion, the analysis performed here does not adequately support this conclusion.

The main factor missing in this analysis is an appropriate accounting for non-emissions-related variability in pollutant concentrations. NO2 concentrations from the 2020 lockdown period were 18% lower than the 5-year mean. This relatively small difference

could easily be explained by meteorological effects, especially since the changes are observed over a short period. From the analysis performed here, it is not possible to say that these decreases are driven by emissions alone. No statistical analysis is performed to indicate that the decrease is significantly lower than what could be expected due to interannual variability. In comparing regions and cities, every observed decrease in pollutant concentration is attributed to emissions changes, while pollutant increases are generally ignored in the discussion. There is no discussion on what may cause the regional differences.

Additionally, the CO analysis consists of comparing TROPOMI observations during the lockdown to the climatology as observed by MOPITT without an evaluation of biases between the instruments. Without this information, there can be no confidence that any decrease seen during lockdown is due to emissions changes and not an artifact of instrument biases. I would not recommend publication of this paper in its current form. However, publication could be warranted if a consistent CO product is used before and after lockdown, a stronger statistical case is made for the significance of the observed changes, and a more thorough discussion of drivers of variability (interannual and regional) that includes impacts of meteorology is included.

Specific Comments: Line 52: CO is not generally considered to be a greenhouse gas itself, although it does contribute to the radiative budget by producing ozone and affecting methane and carbon dioxide oxidation.

Line 57-61: This sentence does not make grammatical sense. Please state which instruments are measuring which pollutant more clearly.

Section 2: Resolution information is incorrect or inconsistent. The resolution of OMI level 3 data is $0.25°x0.25°$, not 25km. MODIS level 3 AOD is $1°x1°$ not 100km. TROPOMI CO is listed as 10 km in the table and 7 km on Line 71. The resolutions of the level 2 CO observation pixels are given, but the data must have been gridded somehow – what resolution (spatial, temporal) was the data gridded to? Was it consistent between MOPITT and TROPOMI? Also, TROPOMI provides high quality NO2 data with finer spatial resolution than OMI – why was OMI chosen for NO2?

Line 80: Some information about the restrictions would be helpful for readers. What was different between phase 1 and phase 2? This is important context in understanding the rest of the results.

Line 88: The authors should take care throughout the paper to distinguish between emissions and concentrations. Emissions were affected by lockdown restrictions, but the data used here represent pollutant concentrations. This mistake is also on Lines 105, 147, 204, 205, 217, 220, 248, 255, 257, and the Table 3 caption.

Figure 1: I find this figure very hard to follow. Could the data processing methodology not be more succinctly described in text?

Figure 2: The text in this figure is very blurry. It is hard to understand what is being shown. The same is true for Figures 4 and 5.

Lines 102: How can a difference be caused by seasonal changes when you are comparing the same time period from two years? More likely this is due to meteorological effects (e.g. winds, cloud cover, etc.).

Line 105: I don't understand what is meant by "detect build up and removal of NO2 during the lockdown period". Maybe this is because the figure is blurry. Do the authors mean that the concentrations increase and decrease during the period? If that is the case, and there is no corresponding change to the lockdown measures, then that's additional evidence of non-emissions factors playing a role.

Line 114: It's not clear why Spain is being mentioned here.

Section 3.1.1: I don't think the analysis here and in Figure 3 show that the NO2 changes are "obviously due" to emissions changes. For one, the lockdown started March 22, and 2020 concentrations are lower than 2019 starting March 7, and lower than 2015-2019 for the whole time period. In Figure 3c, 2020 concentrations in Phase-

2 are higher than in 2017 – essentially, it doesn't look like the concentrations in 2020 in Phase-1 or Phase-2 are outside of what looks like natural interannual variability. A more rigorous statistical approach is needed to demonstrate whether the difference is significant.

Line 137: "there is a decrease in CO levels" – the mean values cited show an increase from 2019 to 2020.

Section 3.2: As mentioned, pre-lockdown CO is from the MOPITT observations and the lockdown values are from TROPOMI. Without evaluating the biases between the instruments, or at least citing research that performs such evaluations, results from this section are not trustworthy. The small changes between 2019 and 2020 could certainly be explained by instrument biases.

Line 139: Assuming that there is no bias between MOPITT and TROPOMI, it's not clear to me why smaller changes in CO compared to NO2 is due to lifetime. If it's an issue of lifetime, how does that explain that in the north and south regions 2020 is higher pre-lockdown, lower in phase 1, and higher again in phase-2? Figure 4d, which is based on TROPOMI alone, seems to show that CO can change quite a bit from week-to-week.

Line 159: Again, I don't see how differences between the same time period in 2019 and 2020 can be due to seasonal differences.

Line 164: "a significant lowering of aerosols. . .is observed over the Indian region during the lockdown" – what about the large increase in the central and eastern regions during Phase 1? Also, the word "significant" usually implies statistical significance, and such statistics were not performed here.

Line 182-184 "A RoC of 5%...during this period" is unclear to me.

Figure 5: Why are the magnitude and distribution of AOD differences in Phase-1 so different than Phase-2, and so different than pre-lockdown? A description of what restrictions were in place would help here.

Table 2 is unclear to me. The text says that colors indicate increases and decreases, so then what do the numbers represent? They do not seem to correspond to each other, although maybe I'm just missing something.

Figure 7: The colors are a bit misleading. For one, darker blue usually indicates a large decrease, and here it means a large increase which may confuse readers. Secondly, the color representing no change (i.e. RoC=0) changes slightly in each row.

Section 3.4: This section could use more discussion. There is quite a bit of difference between regions –some have all pollutants decreasing, some have increases in one pollutant and decreases in another. It would be useful to see at least some speculation about what is causing the regional differences.

Line 257: "Thus, exhaustive lock down. . .led to improvement in the air quality and short term climatic effects over India" overstates the results of the paper. The analysis here shows that some pollutants may have seen decreases in concentration but does not at this point demonstrate that these changes were caused by emissions changes during lockdown and not, for example, meteorology. Secondly, the short term climatic effects are largely speculative and not directly supported by any results here.

---

## Short Comment (SC1) · 15 Jul 2020

This is not a full review of the paper and should not be interpreted as such. It is a comment on the aerosol optical depth (AOD) data source and analysis shown, though these comments may well apply to the other data sets involved too. I was involved in the development of the MODIS Deep Blue aerosol data sets used here. As a minor point, the resolution of these level 3 aggregated data products is not 100 km as Table 1 states but rather 1 degree.

The authors state they are using MODIS Collection 6 data products obtained from the Giovanni visualization website. The latest version (public release began late 2017)

[Figure]

is Collection 6.1. There are some important algorithm and calibration differences between the two versions which may affect the conclusions shown here. The authors should ideally use the latest data versions available. These can be obtained from NASA EarthData search (which is where it looks like the gas data were found) or LAADS directly (https://ladsweb.modaps.eosdis.nasa.gov/). Just make sure to get Collection 6.1. File formats are the same as in Collection 6. See Hsu et al (2019, https://doi.org/10.1029/2018JD029688) for the latest algorithm and Sayer et al (2019, https://doi.org/10.1029/2018JD029598) for the validation. Note that site-by-site AOD validation plots, including for sites shown in this study region, are provided in the Supplement of the Sayer paper.

It was a little hard to read the Figure 5 panel titles to see exactly what is shown, but from the looks of things and the text it seems that (1) pre-lockdown and during-lockdown 2020 and (2) lockdown equivalent 2019 and during-lockdown 2020 were compared? Later there is a comparison to a 5-year (2015-2019) period. It is problematic for the aerosol analysis because we know there is a large amount of seasonal and interannual variability in aerosol loading (driven by changes in sources as well as meteorology and the simple facts of when the satellite overpasses occur). Comparing the periods without the long-term context (as in Figure 5) means you can't attribute the observed AOD changes to lockdown vs. month-to-month or interannual variability or longer-term trends (i.e. "would we have seen this anyway this year").

I will note that MODIS (among other sensors) give a roughly 20-year record of AOD which could be used, plus there are decade or longer accurate and stable AERONET Sun photometers at a number of sites in India, mainly in the IGP. Studies such at Lee et al (2018, https://doi.org/10.3390/rs10091326 ) imply that we need 6 or more years to get a stable (to 0.01) estimate of the annual mean AOD over much of India from monthly composites (their Figure 3); for an estimate of the baseline March-May seasonal AOD climatology they estimates 8-10 years (their Figure A4). This implies that the 5 years used here aren't enough to capture interannual variability within this season.

As a result prominent phrasing such as "Short-term climatological variation of AOD due to lock down" in the Section 3.3 header or the last sentence of the paper, "Thus, exhaustive lock down due to outbreak of COVID-19 has led to improvement in the air quality and short-term climatic effects over India.", worries me since I don't believe we can make an attribution without looking at the broader context or quantifying the uncertainty in the conclusions.

I am also not sure whether change in the mean AOD is the right metric to look at here. We know AOD data are often close to Lognormally-distributed, especially on regional and longer than daily time scales (such as used here). See e.g. O'Neill et al (2000: https://doi.org/10.1029/2000GL011581 ), Sayer and Knobelspiesse (2019: https://doi.org/10.5194/acp-19-15023-2019 ). A change in the mean could mean either that the baseline values are changing, and/or that the magnitudes (and/or frequencies) of extremes in the distributions are changing. The distinction here may be important for e.g. climate or air quality studies. I would encourage the authors to consider which metrics make most sense to report.

The authors mention that the AOD changes might affect the coming monsoon. Given this paper is almost happening in real time, scientifically it would make sense to wait a few months and then see after the fact whether any attributable behaviour in the monsoon was observed. Further, I see that since submission lockdowns have been imposed in India again due (see e.g. https://www.axios.com/india-coronavirus-lockdowns-regions-35febf3b-5480-4d0c-b98b-621d3ecaeb7f.html, selecting this source as it is regarded as not biased and is not behind a paywall). So again a clearer picture might emerge after the pandemic is over and we have the full picture to look at. I don't see the scientific driver behind trying to publish while the event is still going on, given this study isn't to my knowledge being used to guide policy related to the pandemic response.

I expect that similar comments will apply to the trace gas data sets used here as well (depends on sources, lifetimes, transport, variability, and satellite retrieval capabilities),

although am not an expert on those.

This is obviously a topic of great interest at the moment (and I am excited to see studies tackling these questions – though as noted probably it is too soon). I agree that it is seems likely that lockdowns could have had some effect on pollution (including aerosols), but it is quite difficult to quantify that effect. The paper doesn't appear to contain the words "uncertainty", "error", "estimate", or "confidence" or variants of these as far as I can tell (from a text search of the pdf) and any study trying to do an attribution also needs to make a credible attempt to quantify the uncertainty on any change observed.

---

## Author Comment (AC1) · 17 Jul 2020

Dear Dr. Sayer,Thanks for your kind observations and constructive suggestions, which certainly helps us while improving the manuscript.

1.In the present study we have used MODIS level 3 aggregated daily product Âăat 1 degree resolution version 6.1 (MOD08_D3 v6.1 & MYD08_D3 v6.1)). However, it was wrongly written in the current manuscript. We shall correct resolution and product information along with its algorithm development in the revised manuscript. We greatly appreciate the observations and papers suggested for reference.Âă

2. During India's exhaustive lockdown (N=40 days), strict norms had been imposed on all the activities such as industries to shut down, transport sectors, closing of interstate borders and restricting the public movement except for essential needs. After May 4th 2020, lockdown in India has been relaxed in a phased manner. Hence, in the present study we considered only a strict lockdown period, which significantly depicts the influence of anthropogenic activities on air quality. ÂăAs you observed, we compared AOD and other parameters during the lockdown period in 2020 and equivalent period in 2019. Subsequently, we studied the effect of lockdown in 2020 in comparison with the last 5 years, to confirm whether the change is maintained in the last 5 years as well. However, we strongly consider your point because AOD has great variability with sources and local meteorology. We shall update the revised analysis with 10 year data.

3.Also as suggested, we shall explore the other data sources and metrics to improve the result of the study. ÂăWe shall provide all the supporting statistics to strengthens the results reported in the study. Thanks for the references provided.

4. As you are aware aerosols and air pollution together influences cloud properties and contributes to the changes in rainfall. Since India's exhaustive lockdown is just before monsoon, we are only anticipating the forthcoming monsoon may be better than previous year due to probable improvement in air quality. However, now as monsoon started in India, as you suggested we shall consider the supporting evidence and we will report the rainfall during 2019 and 2020. Kindly note India's serious lockdown was upto May 3rd 2020 with serious restriction on every sector and later it was relaxed in phased manner. Thus, this study reported only serious lockdown which makes a benchmark for understanding anthropogenic activities. As you suggested, we shall consider post-lockdown data and we shall compare these results as well in the revised work.

Once again thanks for your valuable comments and suggestions which will definitely help us to strengthen the work.

---

## Referee Comment (RC2) · Anonymous Referee #2 · 16 Aug 2020

This manuscript investigates the changes in NO2, CO, and AOD over the Indian subcontinent while the COVID-19 lockdowns. I appreciate the effort that the authors have undertaken in the preparation of this manuscript. Unfortunately, this manuscript is too disjointed in its current form to recommend publication. The manuscript reads almost like a report without diving deeper into explaining why certain patterns are emerging. Further, some statements are misleading or not scientifically correct. Also, many figures are too blurry to be interpreted, so in some sense, this review is incomplete. Even upon revision, this type of analysis is better for a different journal.

Some specific recommendations for improvement are listed below:

Line 30 - Please include a citation for the specific # of cases

Line 33 - Missing "were" between "production suspended"

Line 36 - Capitalize "China" and please list the names of the three cities.

Line 38 - This entire paragraph is very disjointed and has some factual errors. Some sentences such as the first and ninth are superfluous. And there are some scientific inaccuracies as well. For example, CO is not a greenhouse gas (although some amount will be oxidized to $CO_2$ eventually), ambient air quality monitors measure concentrations not emissions, and there are many sources of biogenic or semi-biogenic sources of aerosols, such as dust, sea salt, volcanic sulfates, secondary organic aerosols, and wildfires (e.g., Siberia), which often overwhelm the anthropogenic signal on a global scale.

Line 68 - The resolutions reported here are not quite correct. Please revise.

Line 74 - TROPOMI also measures $NO_2$; it is unclear why that instrument is not used for the $NO_2$ analysis as well.

Figure 2 - Too blurry to fully evaluate.

Line 102 - Modify "seasonal change" to "meteorological variability between years"

Line 103 - Mentioning the $NO_2$ lifetime is here out of place. Please remove sentence.

Line 114 - Unclear why Spain is mentioned here?

Line 119 - RoC does better than a single year-to-year analysis, but does not account for emissions changes due to economic growth/decline, government policies, or years with anomalous weather patterns. The RoC reduces the effects of meteorology, but it brings up other errors, which are not mentioned. Please modify to state that emission changes due to economic or political factors are not included, and that meteorological factors are somewhat but not entirely removed. This is a very critical point, and must be addressed better throughout the manuscript.

Line 128 - Presumably "LD" is lockdown?

Line 135 - The CO analysis is confusing. When is TROPOMI used vs. when is MOPITT used? Is TROPOMI from 2020 compared to an average of MOPITT from 2015-2019? If so, this is not the scientifically correct manner to conduct this analysis. The two instruments have different spatial resolutions, are ∼20 years different in age, have different overpass times (morning vs. afternoon) and have different algorithms. This section needs to be re-thought. I would recommend using TROPOMI from 2020 and comparing to an average of TROPOMI CO from the prior two yers.

Figure 4 - Too blurry to fully evaluate.

Line 154 - It is confusing as to which AOD products are used here. Terra MODIS is 10:30 local time and Aqua MODIS at 13:30, so it's confusing as to how/why Terra/Aqua is used for one analysis and Aqua only for another analysis. Please double check how the AOD products are being utilized here, and re-calculate as necessary, In particular it's best-practice to not use Giovanni for a scientific paper, and instead download daily L3 files from NASA Earthdata (as has been done for the other products).

Line 156 - Should be Figure 5 not Figure 4

Line 166 - It appears there's an increase in AOD between years between many regions? Am I interpreting Figure 5 correctly? If so, this is glossed over, and should be discussed more in-depth. In general, a much more lengthy discussion is needed here.

Figure 5 - Too blurry to fully evaluate.

Line 181 - Please see comment regarding RoC in response to Line 119. The same comment applies here.

Line 187 - Discussion of the aerosol indirect effect on precipitation is a bit tangential to the theme of the manuscript. I think it would be more appropriate to discuss that higher precipitation can lead to lower aerosol loading irrespective of emissions changes.

Table 2 - I appreciate this table, but it needs to be clarified. There should only be two significant figures when reporting the percentages. Also, it is unclear what the numbers in the brackets are referring to.

Figure 7 - This figure is confusing. It appears that NO2, CO, and AOD are all down before the lockdown period and then are also down after the lockdown period, but there appears to be no effort to account for the pre-lockdown drops in the post-lockdown drops. Therefore any conclusions based on the lockdown periods are misleading.

Figure 8 - This figure is hard to interpret. In particular, it is nearly impossible to line up the cities across the x-axis (time). This type of data is better for a supplemental table. I suggest removal in its current form.

Lastly, there is no discussion of uncertainty throughout the manuscript. This is particularly relevant with respect to quantifying the role of meteorology of the changing trace gas amounts, and in quantifying the role of economic/political impacts on emissions (when comparing 2020 to a 2015-2019 average). Also instrument error for all satellite instruments should be discussed.

---

## Referee Comment (RC3) · Anonymous Referee #3 · 6 Sep 2020

The authors investigate the air pollution trends over India during COVID-19 lock-down period. This is a very interesting topic. The authors have tried to cover multiple species, including NO2, CO2, and AOD. However, I feel in-depth investigations are missing and the writing needs improvement.

General comments: 1. Abstract. "An increase in CO levels was noticeable, probably due to its longer life-time as compared to NO2 and aerosols." The support for this conclusion seems to be missing from the manuscript.

2. Section 3.1. The reason for the increase in NO2 levels is observed in the north, north-east and western parts of India shall be carefully explained.

[Figure]

3. Section 3.2. "During the pre-lockdown period (1-10th March 2020), CO levels were higher as compared to 2019 by ~2%, which is expected due to anthropogenic activities." I suppose 2% change is not a significant change. The reason why this change is driven by anthropogenic activities is also missing.

4. The contribution from natural sources, e.g., biomass burning, is neglect from the whole analysis.

Specific comments:

1. Page 2, line 36. I didn't quite get the logic of putting this sentence here, since it seems not to be associated with the above contents.

2. Page 3, line 71. KNMI and ESA shall be cited when referring to TROPOMI observations.

3. Figure 2. The quality of the figure is not good enough for me to catch the details.

4. Page 4, line 103. I didn't quite get the reason for mentioning the sentence below: "NO2 lifetime is shorter at day time due to reactive photochemical processes in presence of sunlight and longer at night (Richter et al., 2004)."